# p53 activity is selectively licensed in the *Drosophila* stem cell compartment

**Annika Wylie[1][†], Wan-Jin Lu[2][†], Alejandro D'Brot[1], Michael Buszczak[3], John M Abrams[1]\***

[1]Department of Cell Biology, University of Texas Southwestern Medical Center, Dallas, United States; [2]Institute for Stem Cell Biology and Regenerative Medicine, Stanford University School of Medicine, Stanford, United States; [3]Department of Molecular Biology, University of Texas Southwestern Medical Center, Dallas, United States

**Abstract** Oncogenic stress provokes tumor suppression by p53 but the extent to which this regulatory axis is conserved remains unknown. Using a biosensor to visualize p53 action, we find that *Drosophila* p53 is selectively active in gonadal stem cells after exposure to stressors that destabilize the genome. Similar p53 activity occurred in hyperplastic growths that were triggered either by the Ras[V12] oncoprotein or by failed differentiation programs. In a model of transient sterility, p53 was required for the recovery of fertility after stress, and entry into the cell cycle was delayed in p53[-] stem cells. Together, these observations establish that the stem cell compartment of the *Drosophila* germline is selectively licensed for stress-induced activation of the p53 regulatory network. Furthermore, the findings uncover ancestral links between p53 and aberrant proliferation that are independent of DNA breaks and predate evolution of the ARF/Mdm2 axis.

**\*For correspondence:** John.
Abrams@utsouthwestern.edu

[†]These authors contributed
equally to this work

**Competing interests:** The
authors declare that no
competing interests exist.

**Reviewing editor**: Carol Prives,
Columbia University, United
States

## Introduction

Throughout the animal kingdom, p53 occupies a central position within conserved stress response networks. The protein integrates diverse signals associated with DNA damage and uncontrolled proliferation to govern adaptive downstream responses such as increased DNA repair, arrested cell cycle, and apoptosis (*Vousden and Lane, 2007*). Where examined, the genes encoding p53 are not essential for viability but have been implicated as regulators of aging (*Derry et al., 2007*; *Donehower et al., 1992*; *Lee et al., 2003*; *Sogame et al., 2003*). It is now well appreciated that ancestral roles for this gene family must have predated functions in tumor suppression. In support of this, members of the p53 gene family are present in unicellular protists and short-lived multicellular organisms (*Lu et al., 2009*; *Mendoza et al., 2003*; *Nordstrom and Abrams, 2000*). Furthermore, cancer was probably a negligible source of selection pressure during the course of human evolution (*Aranda-Anzaldo and Dent, 2007*) and the combined removal of canonical p53 effectors (p21, Puma, and Noxa) does not account for tumor suppression in mice (*Valente et al., 2013*). These and other observations suggest that tumor suppressive roles for the p53 family were co-opted from primordial functions, some of which may have been linked to meiotic recombination (*Lu et al., 2010*).

In recent years, considerable evidence has surfaced linking p53 action to stem cell biology. For example, in mammary stem cells p53 promotes asymmetric division and cell polarity, thereby helping to limit the population of stem cells in the mammary gland (*Cicalese et al., 2009*). Furthermore, reprogramming of somatic cells into induced pluripotent stem cells (iPSCs) is greatly increased in p53 deficient cells, suggesting that p53 may act as a 'barrier for induced pluripotency' (*Krizhanovsky and Lowe, 2009*). Consistent with this, several labs have shown that p53 induces embryonic stem cell differentiation to maintain genomic stability after DNA damage (*Lin et al., 2005*; *Neveu et al., 2010*;

**eLife digest** The most common genetic change seen in cancer patients produces a faulty version of the p53 protein, which normally restricts tissue growth. This change promotes cancer because cells can now divide faster and fail to die when they should. Much remains to be learned about how p53 functions to restrain growth. As p53 is found in primitive organisms, and cancer is unlikely to have significantly influenced evolution, suppressing tumor formation was almost certainly not the original function of this gene. Furthermore, p53 works in a different way compared to many other tumour suppressors. Therefore, prevention of cancer is likely to have evolved as a side effect derived from more ancient functions.

Recently, a link between p53 and stem cells has been uncovered. Stem cells are special because they can develop into many different types of cells, and they are crucial for the growth and repair of tissues. To form a particular type of cell, the stem cell divides to create two daughter cells. Commonly, one daughter cell stays in the stem state, whereas the other becomes a particular type of cell, such as a nerve cell or muscle cell. Because of this special property, scientists hypothesize that stem cells have special mechanisms to protect them from DNA damage that might partially depend on p53. This would prevent the spread of damaged genomes that would otherwise occur among daughter cells.

To learn more about how p53 influences stem cells, Wylie, Lu et al. monitored its activity in the gonads of fruit flies, which are a powerful genetic model. They found that damaging DNA activates p53 in stem cells and their daughter cells, but not in other types of cells that have been damaged. In addition, p53 is activated by the uncontrolled growth and division of stem cells in the gonad, even when DNA is not damaged. This is unexpected since molecules linking inappropriate growth to p53 were thought to be present only in mammals. Therefore, it appears that the tumor-suppressing behavior of p53 in mammals was adapted from its more ancient ability to regulate stem cell growth—an ability that evolved before organisms divided into vertebrates and invertebrates.

*Zhao and Xu, 2010*). Together with recent studies in planaria, these observations indicate that an ancestral focus of p53 action could operate in stem cells (*Pearson and Sanchez Alvarado, 2009*). We directly tested this possibility using a p53 biosensor to visualize *Drosophila* germline stem cells and their progeny. When DNA breaks were exogenously imposed or intrinsically engineered, *Drosophila* p53 (Dp53) was activated selectively in germline stem cells (GSCs) and their immediate daughters, indicating that these cells are uniquely licensed for p53 action. Furthermore, in various germline tumor models Dp53 was constitutively hyperactivated, suggesting that ancient links between p53 and inappropriate growth predate canonical effectors that connect these regulatory networks (e.g., ARF and MDM2).

## Results

### Damage-induced Dp53 activity in the germline is restricted to stem cells

The *Drosophila* gonad is a classic system for studying the stem cell compartment since stem cells, their immediate daughters, and the surrounding niche are easily identified. In the ovary, germline stem cells (GSCs) undergo self-renewing divisions that typically produce a GSC and a cystoblast (CB). These GSCs support egg production throughout the lifespan of female adults (*Figure 1B*). We used in vivo biosensors (*Lu et al., 2010*; *Brodsky et al., 2000*) to visualize p53 activity as GSCs responded to various sources of stress (*Figure 1A*). To exclude technical artifacts, two GFP reporters were used—one localizes to the nucleus (p53R-GFPnls) and the other does not (p53R-GFPcyt). As previously described (*Lu et al., 2010*), programed p53 activity triggered by meiosis was only observed in region 2 (*Figure 1B*). After exposure to ionizing radiation (IR) stress, p53 activity was induced in virtually all germaria. However, despite widespread damage to the organ (*Figure 1—figure supplement 1*), this unprogrammed response was remarkably restricted to germline stem cells (GSCs) and their immediate progeny (CBs) (*Figure 1C,E*). Furthermore, as seen in *Figure 1—source data 1A*, this response was highly penetrant. Since we rarely observe reporter activation only in CBs, the signal seen in CBs probably reflects GFP perduring from the parental stem cells. Furthermore, post-irradiation levels of

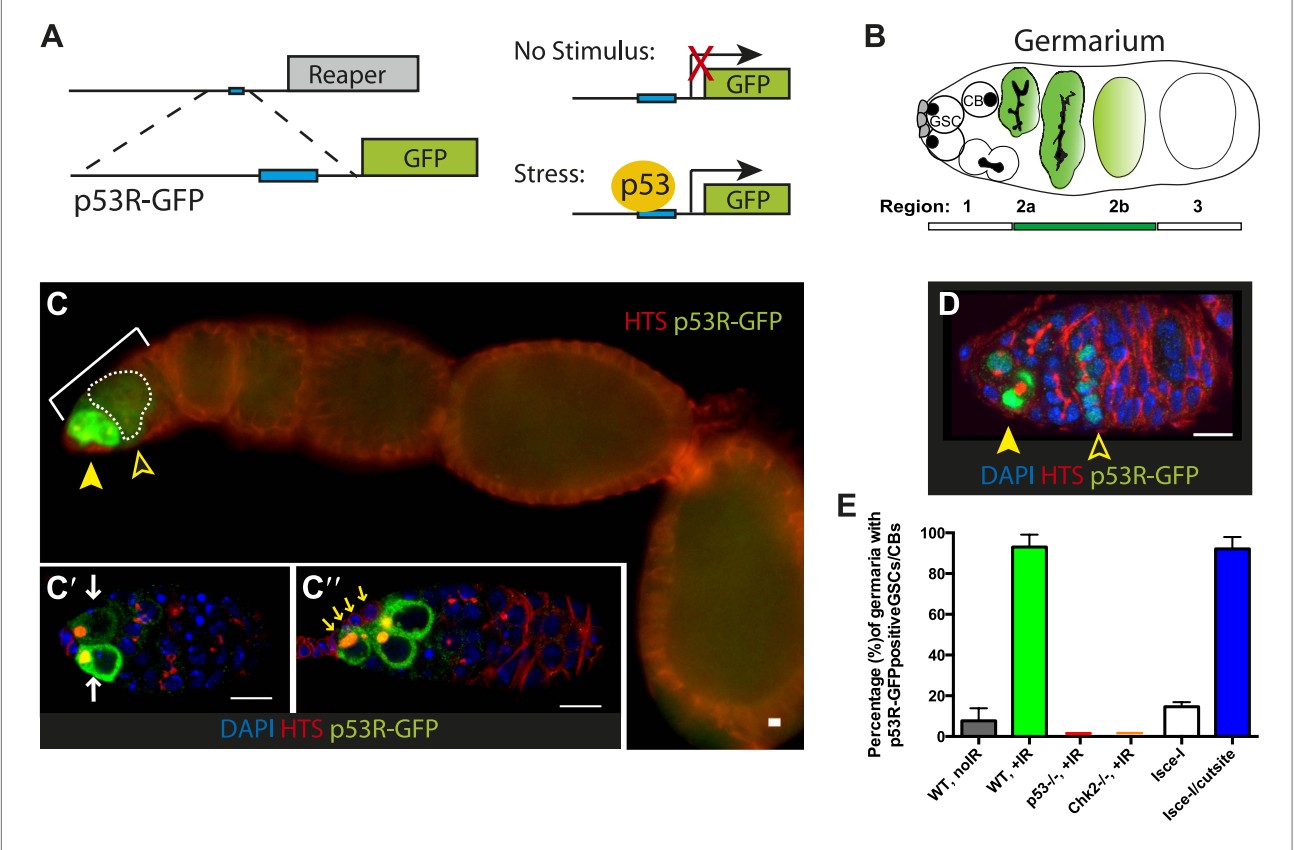

**Figure 1**. Genotoxic stress selectively triggers p53 activity in ovarian stem cells. (**A**) Construction of p53 biosensors. A well-characterized p53 enhancer (black line) that contains a p53 consensus binding site (blue box) conserved from flies to humans resides upstream of the *reaper* locus (gray box) (***Brodsky et al., 2000***). A 150-bp fragment containing this enhancer was placed upstream of GFP (p53R-GFP). Transgenic fly strains are made with two reporter constructs, one contains a nuclear localization signal for GFP (p53R-GFPnls) and the other one does not (p53R-GFPcyt). Stimuli that trigger p53 activation induce GFP expression. These biosensors require wild-type p53 and are effective readouts for p53 function. (**B**) Germline stem cells (GSCs) are in contact with cap cells (in gray) at the apical tip of the germarium and undergo self-renewing division to produce a GSC and cystoblast (CBs) (***Spradling et al., 2001***). In unperturbed ovaries, programed activation of the p53R-GFP biosensor is triggered by meiotic recombination in region 2 of the germarium, marked by open arrowhead in (**C**) and (**D**) (***Lu et al., 2010***). (**C**) After radiation challenge (IR) the p53R-GFPcyt biosensor (green) is selectively induced in ovarian GSCs and CBs noted by a solid arrowhead. Bracket denotes the germarium. The open arrowhead and dotted line indicates p53 activation in region 2 prompted by meiosis. Insets (**C'** and **C''**) are confocal images from different irradiated germaria counterstained with DAPI (blue). p53R-GFPcyt induction (green) initiates in GSCs that exhibit rounded fusomes (**C'** white arrows) labeled by α-HTS (Hu li tai shao, red) and are in contact with cap cells (**C''** yellow arrows). Cells that activate p53 in (**C'** and **C''**) were confirmed to be germ cells by α-Vasa staining (shown in ***Figure 1—figure supplement 2C–D'***). (**D**) An engineered DNA double-stranded break (DSB) mediated by I-SceI (see texts and 'Materials and methods') induces the p53R-GFPnls biosensor (green) in GSCs/CBs, noted by a solid arrow. Open arrow indicates meiotic p53. The germarium is counterstained with α-HTS (red) and DAPI (blue). (**E**) Quantifies the percentage of germaria activated for the p53 biosensors in GSCs and their immediate progeny. Note that the perturbation-dependent responses reported here are all highly penetrant. Selective activation is IR (green) and I-SceI (blue) dependent at the 0.001 significance level. Note that biosensor activation did not occur in *p53*[−/−] (red) or *chk2*[−/−] (orange) mutants (see ***Figure 1—figure supplement 2A',A''***). Sample sizes are combined from at least two independent trials (available in ***Figure 1—source data 1***). All scale bars represent 10 μm. In panels **C–C''** the p53R-GFPcyt reporter was used. In panel **D**, the p53R-GFPnls biosensor was used.

The following source data and figure supplements are available for figure 1:

**Source data 1**. Validation of the p53R-GFP biosensors.

**Figure supplement 1**. Wide-spread DNA breaks after irradiation.

**Figure supplement 2**. Selective p53 action in germline stem cells is detected using a p53 biosensor.

**Figure supplement 3**. ATR is not rate limiting for p53 activation in the germline.

GFP were noticeably more robust than the programed activity during meiosis (compare solid arrows to open arrows in *Figure 1C,D*) (*Lu et al., 2010*). As expected, p53 biosensor activity was not observed within the ovary of p53$^{-/-}$ animals and was also absent from ovaries lacking the upstream Chk2 kinase (*Figure 1E*, *Figure 1—figure supplement 2A',A''*, *Figure 1—source data 1A*).

Double stranded DNA breaks (DSBs) are responsible for many of the biological effects associated with IR (*Ward, 1994*). Therefore, to determine whether DSBs are sufficient to induce the p53 reporter, we ubiquitously expressed the I-SceI endonuclease in the germline of flies engineered to harbor a single I-SceI recognition site in each nucleus. As seen with IR exposure, p53 activity occurred only in GSCs/CBs when DSBs were induced (*Figure 1D,E*, *Figure 1—figure supplement 2B*, *Figure 1—source data 1B*). Furthermore, it is notable that a single DSB was sufficient to provoke robust p53 activity in GSCs/CBs. Therefore, whether exogenously imposed or intrinsically engineered, DSBs triggered p53 selective activation that was confined to GSCs and their immediate progeny. Furthermore, this stem cell restricted response is clearly under genetic control. For example, in directed tests of chosen mutants we identified a class of lesions that exhibit non-selective p53 action throughout the ovary only after IR challenge (see *Figure 1—figure supplement 3*, *Figure 1—source data 1C*). Therefore, p53 is present and potentially functional in all cells of the ovary but, under normal conditions, its action is somehow confined to GSCs and their immediate progeny.

To ask whether this pattern might reflect a general property of germline stem cells, we similarly examined the male gonad. As seen in the ovary, we observed selective p53 reporter activation in GSCs and their immediate progeny (gonioblasts) in irradiated testis (*Figure 2*). Likewise, stimulus-dependent activity required p53 and was not seen in unchallenged testis (*Figure 2C*, *Figure 2—figure supplement 1A,B*, *Figure 1—source data 1A*). Occasionally, the biosensor was also present in early spermatogonial cysts, perhaps reflecting perduring GFP and/or independent activation associated with dying cells (*Figure 2D–D''*, *Figure 2—figure supplement 1D-E''*). Collectively, the observations in *Figure 1* and *Figure 2* demonstrate that selective p53 activation in the stem cell compartment is a general property of germline tissues exposed to genotoxic stress. We note that perturbation-dependent induction of the p53 biosensor in gonadal stem cells was highly penetrant (*Figure 1E*, *Figure 2C*). However, like all stress responses, the strength of signal and the number of responding cells were variable from animal to animal (*Figure 1C,D*, *Figure 2D*) perhaps reflecting distinct cell cycle dynamics occurring in GSCs at the time of challenge.

## Genome instability provokes p53 action in the stem cells

We tested whether other genome destabilizing factors elicited similar p53 activity in stem cells. To examine the effect of deregulated retrotransposons, we introduced the p53 biosensor into *cutoff* or *aubergine* mutant animals. These genes encode essential components of the piwi-associated RNA (piRNA) pathway, acting to silence retrotransposons in the germline (*Chen et al., 2007*). The corresponding mutants exhibit disregulated retrotransposition, reduced fecundity, and egg shell ventralization (*Chen et al., 2007*). *Figure 3A* shows that in *cutoff* mutants induction of p53R-GFP occurs exclusively in GSCs and their progeny at a penetrance comparable to irradiated wild-type animals (*Figure 3—source data 1*). Frequent p53 activation in the germline was similarly observed in the GSCs of *aubergine* mutants (*Figure 3B*) and *rad54* mutants defective for DNA repair (*Figure 3C*). However, in contrast to *cutoff* mutants, the p53 biosensor was not entirely restricted to GSCs/CBs in these mutants (*Figure 3—figure supplement 1B,C*, *Figure 3—source data 1*) perhaps reflecting differences in the kinetics of repair that may occur in these different backgrounds (*Klattenhoff et al., 2007*).

## p53 enables recovery from stress-induced sterility and proper exit from proliferative arrest

In somatic cells, Dp53 promotes stress-induced apoptosis (*Sogame et al., 2003*). Therefore, we examined the germarium for evidence of cell death by detecting cleaved caspase-3. In the 24-hr period post challenge, over 90% of GSCs induce the reporter but the average incidence of apoptosis was less than 4% (*Figure 4—figure supplement 1*, *Figure 4—source data 1A*). Furthermore, we did not observe an obvious role for p53 in regulating stem cell numbers in the *Drosophila* ovary in the presence or absence of stress (*Figure 1—source data 1*, *Figure 4—source data 1A*). We also used α-pH2Av immunostaining, the *Drosophila* counterpart of mammalian pH2AX (*Mehrotra and McKim, 2006*), to follow the repair of DSBs after IR and found that resolution of these lesions was unaffected in the germaria of p53 mutants (*Figure 4—figure supplement 2*). Similarly, in BrdU incorporation

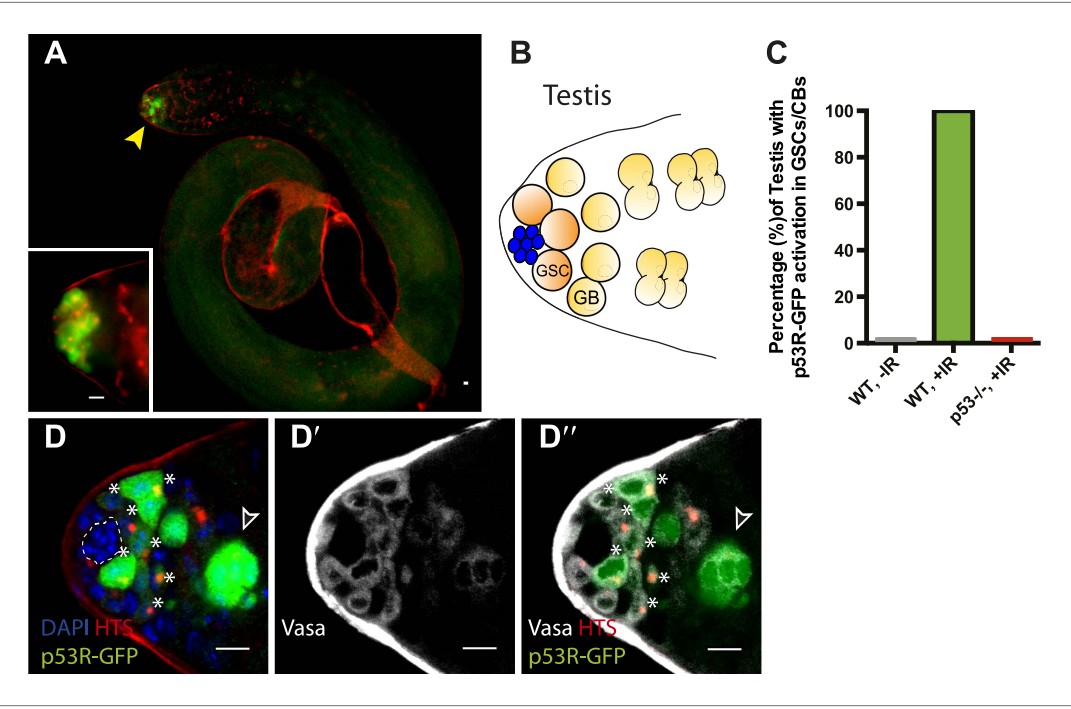

**Figure 2**. Selective p53 activity occurs in male germline stem cells. (**A**) p53R-GFPcyt (green) is induced at the apical tip of an irradiated testis (arrowhead), where stem cells are located (see **B**). α-HTS co-staining (red) highlights early stages of germline development. The inset in panel (**A**) shows a higher magnification view from a different irradiated testis. (**B**) Male GSCs are in contact with cap cells (blue flower pattern) at the apical tip of the testis and divide to produce a gonioblast daugther (GB). (**C**) Quantifies the percentage of testis activated for the p53 biosensors in GSCs and their immediate progeny. Selective activation is IR (green) dependent and conditional upon p53 since p53R-GFP activation did not occur in *p53⁻/⁻* mutants (red bar). (**D–D''**) Confocal images from other irradiated testes confirmed that stem cells induced for p53R-GFPcyt (green, **D** and **D''**) are also positive for rounded HTS staining (red, **D** and **D''**) and the germline specific marker Vasa (white, **D'** and **D''**) as expected. The hub (dotted line, **D**) was routinely identified by the characteristic nuclei pattern as illustrated in **B** (blue cells) and by negative Vasa staining (**D'** and **D''**). Asterisks mark p53R-GFP positive cells that are adjacent to the hub and Vasa positive or Vasa positive with rounded fusomes. Also note that the hub was identified by α-Armadillo staining (*Figure 2—figure supplement 1C*). Open arrowhead in (**D** and **D''**) is likely a dying cyst as indicated by pyknotic and condensing nuclei and irregular HTS (*Figure 2—figure supplement 1D–E*). In panels **A**, **D–D''** the p53R-GFPcyt reporter was used. All scale bars represent 10 µm.

The following figure supplements are available for figure 2:

**Figure supplement 1**. p53 Reporter activation in the male germline (seen in *Figure 2A*) is conditional upon irradiation (A) and is p53 dependent (B).

studies, the rates at which wild-type and p53⁻/⁻ GSCs/CBs entered proliferative arrest were also indistinguishable (*Figure 4A*). However, in the post-stress period, we did observe that p53 mutants were significantly delayed for re-entry into the cell cycle (*Figure 4A*). Furthermore, this defect is reversed in p53 genomic rescue strains confirming an assignment of this phenotype to the p53 locus (*Figure 4—source data 1B*).

To examine how the action of p53 might coordinate adaptive stress responses in GSCs, we developed a fertility recovery assay. In this study, females were irradiated to induce transient sterility and the recovery of fertility was scored over time (see 'Materials and methods'). *Figure 4B* shows that wild-type females recovered from infertility within 1 week post-exposure to IR at a dose of 11.5 krad. In contrast, females lacking p53 remained permanently infertile even when tracked over 2 weeks after IR (*Figure 4B*). To confirm that p53 gene function is responsible for this phenotype, we tested p53⁻ females carrying a genomic rescue fragment spanning the p53 gene (see 'Materials and methods'). We tested two rescue strains and in both cases the sterility defect was reversed (*Figure 4B*).

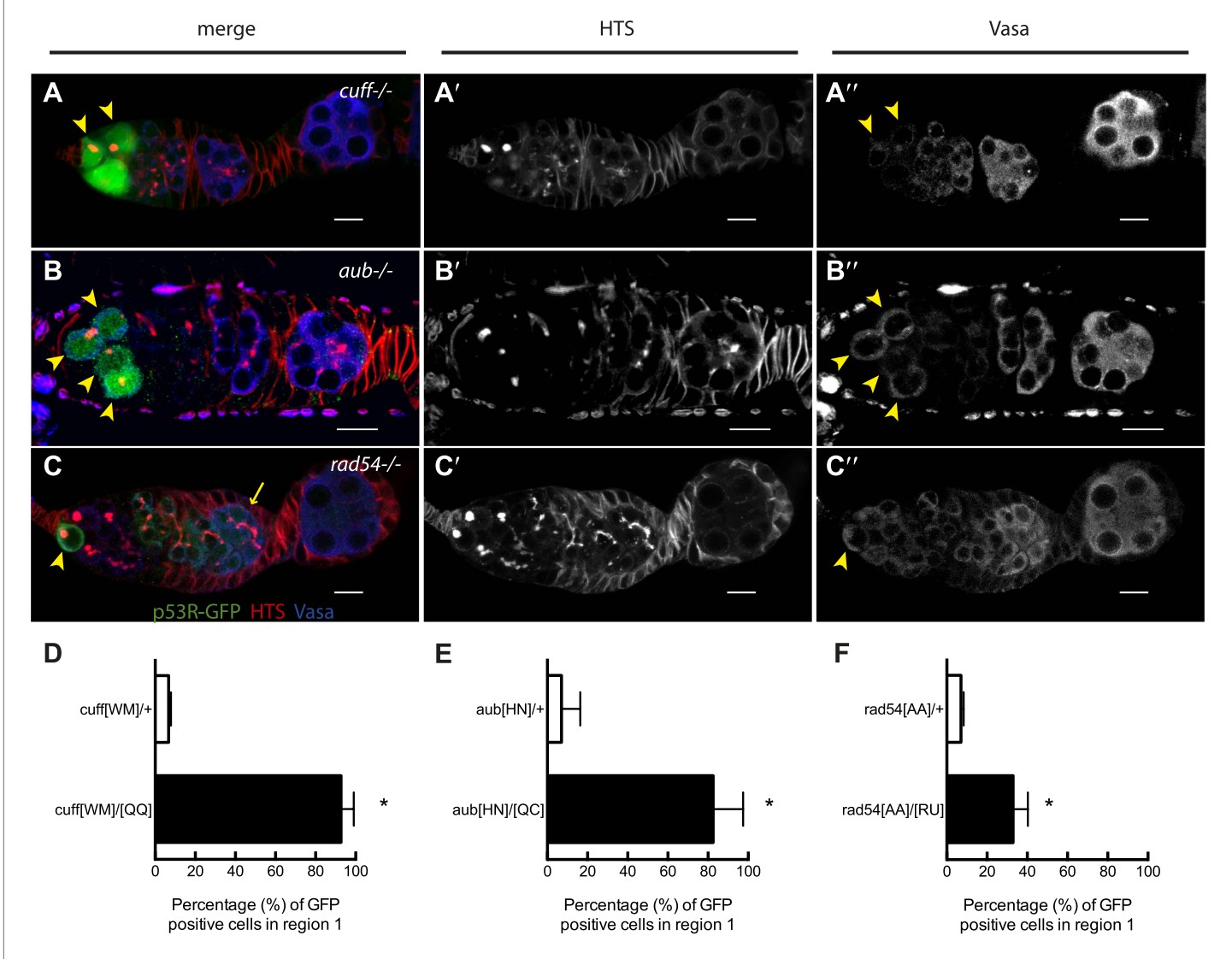

**Figure 3**. Stem cell associated p53 activity in defective DNA repair and retrotransposon silencing mutants. (**A–B**) Activation of the p53 biosensor (green) in the germarium of piRNA mutants, (**A**) *cutoff*[QQ]/[WM] and (**B**) *aubergine*[HN]/[QC]. (**C**) Activation of the p53 biosensor in *rad54*, a meiotic DNA repair mutant. (**D–F**) Germaria were found to express p53R-GFPcyt in GSCs/CBs with a penetrance of 90% for *cutoff* mutants (**D**, p<0.0001), 80% for *aubergine* mutants (**E**, p=0.0018), and 33% for *rad54* mutants (**F**, p=0.0039). Asterisks indicate significant differences between heterozygous controls and homozygous mutants. GSCs/CBs were identified by rounded fusomes detected with α-HTS (red in merge **A**, **B**, **C** and white in **A'**, **B'**, **C'**). Arrowheads indicate that p53R-GFP positive cells are also germ cells identified by Vasa staining (blue in **A**, **B**, **C** and white split channel in **A''**, **B''**, **C''**). Note that this particular α-Vasa antibody cross-reacts against the muscle sheath that surrounds each ovariole. If the sheath is not fully dissected and removed, then background staining is evident, as seen in *Figure 2B*''. Control genotypes were *cuff*[WM]/CyO, *aub*[HN]/CyO, *rad54*[AA]/CyO. Note that *aub* and *rad54* mutants occasionally showed p53 activation beyond region 2 of the germarium (arrow in **C**), quantified in *Figure 3—figure supplement 1*, *Figure 3—source data 1*. All scale bars represent 10 μm. In panels **A**, **B**, and **C**, the p53R-GFPcyt reporter was used.

The following source data and figure supplements are available for figure 3:

**Source data 1**. Quantification of p53 activation in defective DNA repair and retrotransposon silencing mutants.

**Figure supplement 1**. Quantification of p53 activation in defective DNA repair and retrotransposon silencing mutants in region 3 and stage 2–8 egg chambers.

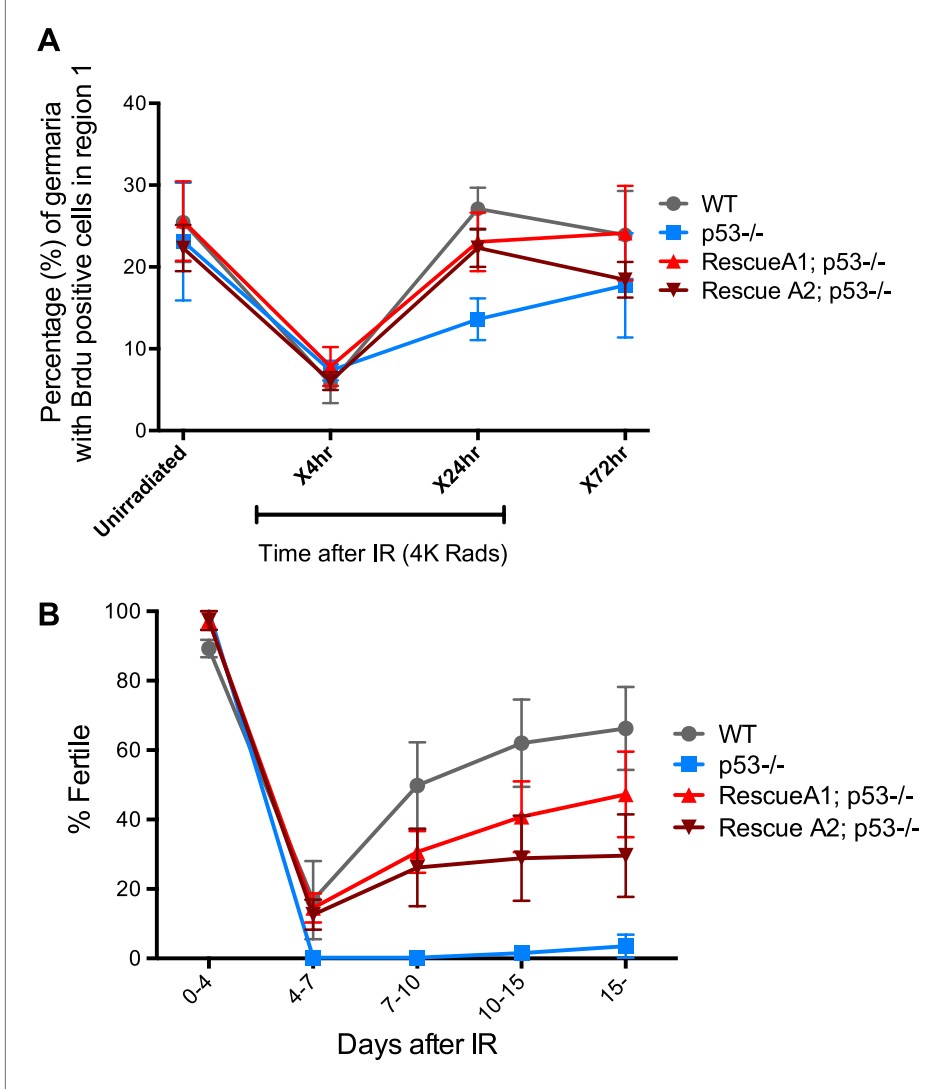

**Figure 4**. p53 mutants exhibit impaired fertility and delayed re-entry into the cell cycle after irradiation. (**A**) BrdU incorporation in GSCs after 4 krad of IR. The percentage of germaria containing BrdU positive GSCs/CBs was plotted on the Y axis. WT and p53$^{-/-}$ GSCs arrest with similar kinetics but p53$^{-/-}$ GSCs were significantly delayed for re-entry into the cell cycle. Error bars represent standard deviation from tests of three independent cohorts. WT and two rescue strains are significantly different from p53$^{-/-}$ at the 0.05 level at the x24 hr time point. Percentages and number of germaria assayed are included in **Figure 4—source data 1B**. In panels **A** and **B**, p53$^{-/-}$ represents animal transheterozygous for two p53 null alleles, p53$^{ns}$ and p53$^{K1}$. (**B**) Fertility in wild-type (WT) and p53$^{-/-}$ females was measured after exposure to 11.5 krad of IR (see 'Materials and methods'), which induces persisting sterility in p53 mutants. WT fertility is significantly different from p53$^{-/-}$ during time points 7–10, 10–15, and 15- at the 0.05 level (see 'Materials and methods'). Two rescue strains showed partial restoration of fertility. Rescue 1A strain showed restored fertility is significantly different from p53$^{-/-}$ at the 0.05 level at days 10–15 and 15-. Note that after 15 days post irradiaton, fertility was monitored for at least 9 more days as indicated by 15-. Error bars represent standard deviation from five independent trials.

The following source data and figure supplements are available for figure 4:

**Source data 1**. Quantification of proliferative potential and apoptosis of germaria challenged with irradiation.

**Figure supplement 1**. Reporter activation after irradiation does not lead to purging of GSCs through apoptosis.

*Figure 4. Continued on next page*

*Figure 4. Continued*

**Figure supplement 2**. Radiation-induced DNA double-stranded breaks appear and disappear with similar kinetics in WT and p53$^{-/-}$ GSCs.

**Figure supplement 3**. Fertility recovery correlates with proliferation by GSCs and their progeny.

However, neither rescue strain fully restored fertility to wild type levels, possibly reflecting incomplete restoration of wild type regulation in the transgenes.

To test whether we could link the fertility defect (*Figure 4B*) to the cell cycle defects observed at a lower dose (*Figure 4A*), we examined fertility and cell cycle kinetics at an intermediate dose (9 krad) of IR. After this challenge, p53$^{-/-}$ females exhibit impaired fertility, whereas WT flies remained fertile (*Figure 4—figure supplement 3*). We performed BrdU incorporation studies over 7 days with females irradiated at 9 krad and assayed the number of germaria that had BrdU positive cells in region 1. Under these conditions, we observed persistently reduced proliferative activity in p53$^{-/-}$ stem cells even 7 days after IR (*Figure 4—figure supplement 3C*, *Figure 4—source data 1C*). This result is consistent with the possibility that fertility defects seen in p53$^{-/-}$ flies are linked to the impaired cell cycle kinetics found in GSCs. Furthermore, the data in *Figure 4B* and *Figure 4—figure supplement 3A* suggest that radiosensitivity associated with the p53$^{-/-}$ genotype, previously been documented for larval stages (*Sogame et al., 2003*), also applies to germline tissue.

## Uncontrolled stem cell proliferation activates p53

Oncogenic properties are thought to simulate 'stemness' and oncogenic signals frequently result in p53 activation (*Vousden and Lane, 2007*). However, it is not known whether this regulatory axis is conserved beyond mammals. To test whether inappropriate growth triggers *Drosophila* p53 function, we examined the p53 biosensors in various germline tumor models. First, we expressed an oncogenic form of RAS commonly found in human cancers together with the p53 biosensor (*Lee et al., 1996*). Transient expression of the *Drosophila* Ras$^{V12}$ counterpart provoked robust p53 activation mainly in the GSCs and CBs (*Figure 5B*, *Figure 5—source data 1*). *Figure 5E* shows that another oncoprotein, Cyclin E, produced similar results. We also examined these biosensors in *bam* mutants, where a block in differentiation causes extensive hyperplasia (*McKearin and Ohlstein, 1995*) and in these tumors extensive reporter activity was also seen (*Figure 5C,D*). Likewise, expanded BMP (bone morphogenic protein) signaling (*Chen and McKearin, 2003*) or reduced *Lsd1* (lysine-specific demethylase 1) activity (*Eliazer et al., 2011*) in neighboring somatic cells can also cause inappropriate growth and robust p53 activity was similarly observed in these germline tumors as well (*Figure 5F,G,H*). Therefore, whether caused by forced oncoprotein expression (panels B, E), failed differentiation programs (panels C, D), or expansion of the stem cell niche (panels F–H), inappropriate growth of *Drosophila* tissues was consistently accompanied by p53 activity. As seen with genotoxic stress, biosensor responses seen in these contexts was somewhat variable, perhaps reflecting complex signaling and/or cell cycle dynamics that occur in these tumor models. Technical sources of variation linked the UAS-GAL4 driver system and/or non-uniform accumulation of the oncogenic product could also contribute to variability in these contexts.

We considered the possibility that inappropriate growth might indirectly activate p53 by provoking DNA damage. To test this, we stained *bamΔ$^{86}$* ovaries for pH2Av (*Joyce et al., 2011*). We observed very few pH2Av foci in *bam* tumors and, notably, these foci did not co-localize with p53 biosensor activity (*Figure 5—figure supplement 1*). Therefore, p53 activity in these tumors is not triggered by DSBs but instead, appears to be directly triggered by signals associated with hyperplastic growth.

As seen in *Figure 5*, diverse types of hyperplastic growth triggered constitutive p53 activity. To ask how p53 functions in these tumors, we examined bam$^{-/-}$ ovaries that were either WT or null for p53. Tumor size was not significantly altered in the absence of p53, but we did observe dramatically altered cytology in tumors that lacked p53. As seen in *Figure 6A*, bam$^{-/-}$ ovarian cysts are typically filled with stem-like cells that exhibit round or dumbbell-shaped fusomes when stained with α-HTS (*Lin et al., 1994*). As documented in *Figure 6C*, defective fusomes were seen in all bam$^{-/-}$;p53$^{-/-}$ cysts and, in nearly half of these unusually large nuclei were observed. Though not quantified, micronuclei were also prevalent in these samples. Since defective fusome morphologies and irregular nuclei are

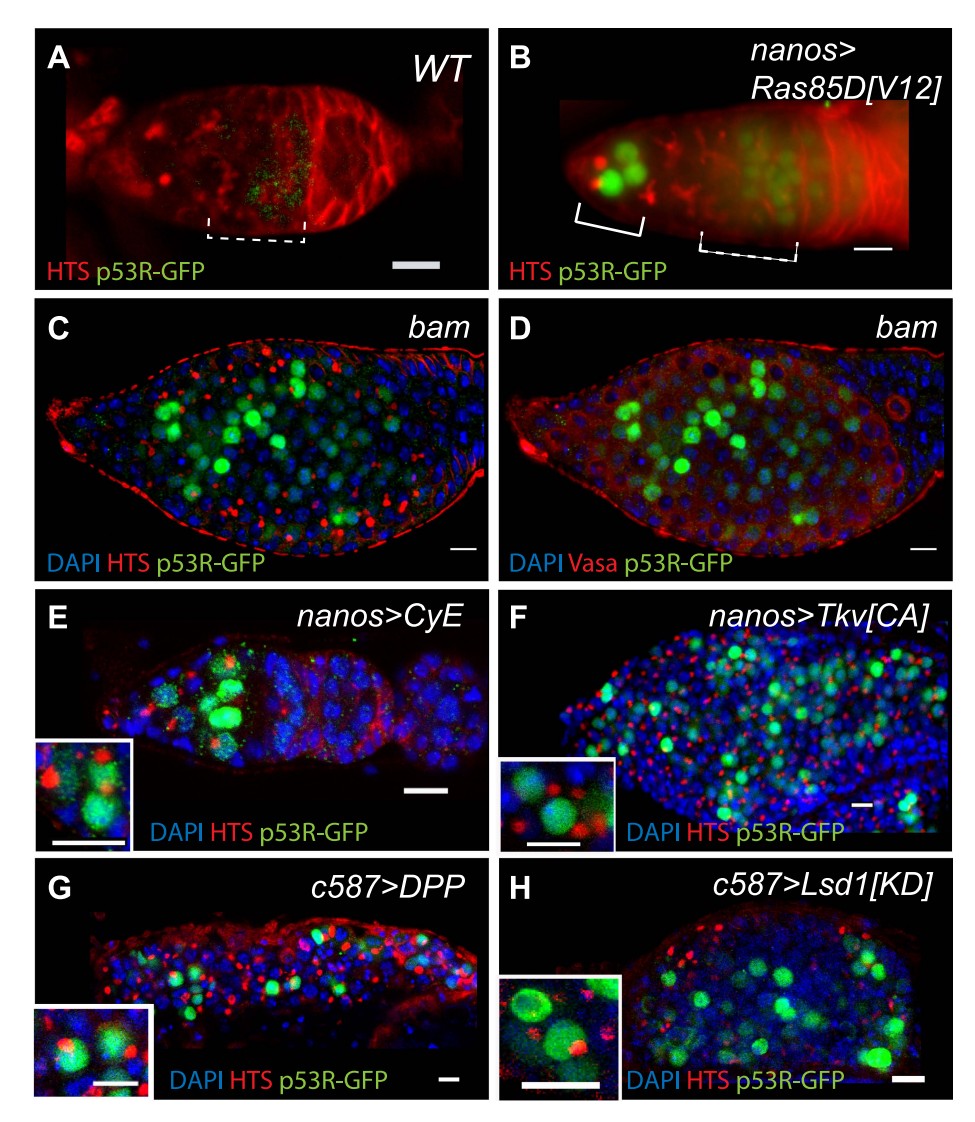

**Figure 5**. Deregulated growth in the stem cell compartment provokes p53 action. (**A**) In an unperturbed wild-type (WT) germarium, the p53R-GFPnls biosensor is absent from GSCs/CBs, marked here by rounded fusomes stained with α-HTS (red). The modest signal in region 2 reflects meiotic p53 activity (dotted bracket) (*Lu et al., 2010*). When perturbed by Ras[V12] (**B**) the p53 biosensor (green) is induced in GSCs/CBs (solid bracket in **B**, see *Figure 5—source data 1*). Perturbation from failed differentiation programs caused by the *bam* mutation (**C–D**) or Cyclin E over-expression (**E**) provokes similar p53 biosensor activity. Likewise, increased DPP signaling caused by a constitutively active Tkv receptor (**F**) or ectopic DPP ligand expression (**G**) also prompts induction of the p53 reporter. Induction of the p53 reporter is also seen, when the stem cell niche is expanded by silencing of *Lsd1* (**H**) (*Eliazer et al., 2011*). Insets in panels **E–H** are magnified views of tumor cysts showing that p53R-GFP positive cells exhibit stem-like properties with rounded fusomes detected by α-HTS co-staining (red). Note in panels **B**, **E** and **F**, the indicated UAS transgenes were expressed using the germline specific driver, nanos-GAL4VP16 (*Rorth, 1998*). For panels **G** and **H**, expression was achieved by the driver c587-GAL4 in somatic cells of the ovariole tip (*Song et al., 2004*). All images shown are immunostainings for the p53R-GFPnls biosensor (green), HTS (red), and/or DAPI (blue) except for panel **D** which was co-stained with α-Vasa (red) to show that p53 activated cells retain the germline marker in *bam* mutants. All other panels (**A–C, E–H**) were stained with α-HTS (red). Note that panel **D** stained with α-Vasa is the same *bam* ovariole shown in **C** with α-HTS. Relevant quantification including the nanosGAL4 driver alone is shown in *Figure 5—source data 1*. Scale bars = 10 µm.

*Figure 5. Continued on next page*

*Figure 5. Continued*

The following source data and figure supplements are available for figure 5:

**Source data 1**. Quantification of biosensor activity in germline tumors.
**Figure supplement 1**. Reporter induction during forced proliferation signals is independent of DNA damage.

consistent with aberrant mitosis, our data suggests a role for p53 in promoting proper cell cycle progression in these stem-like tumors.

To further examine the functional role of p53 in this context, we examined gene expression profiles of bam$^{-/-}$ ovaries that were WT or null for p53 by microarray. In total, we found that 297 gene transcripts were altered by at least twofold or greater in the absence of p53. *Table 1* lists the top 20 genes that are affected (upregulated or downregulated) by p53 in these tumors. Using the Gene Expression Commons (GEXC) tool, we compared these gene sets to existing germline, embryonic, and somatic expression profiles. We did not find a coherent pattern among the top 20 genes that are normally upregulated by p53. However, among the top 20 genes that are normally suppressed by p53 in these germline tumors, we observed a modest enrichment for transcripts that were absent in either the embryonic stages or other somatic tissues (*Table 1—source data 1*). These data, together with our histological studies (*Figure 6*), establish that p53 exerts functional activities that impact cellular and molecular properties of *Drosophila* stem cell tumors.

## Discussion

We found that adult *Drosophila* exposed to genotoxic stress or genome destabilizers selectively activated p53 in GSCs and their immediate progeny. This striking specificity was observed despite widespread Dp53 expression (*FBgn0039044 m.*; *Jin et al., 2000*; *Ollmann et al., 2000*) and widespread

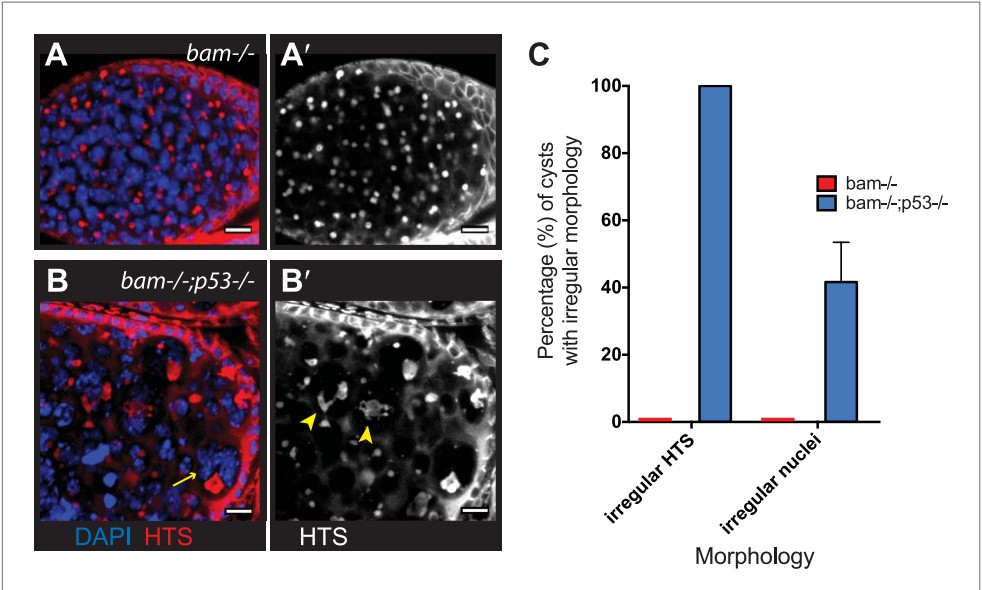

**Figure 6**. Abnormal fusomes and irregular nuclei are seen in bam$^{-/-}$p53$^{-/-}$ tumors. (**A–A'**) Cells in bam$^{-/-}$ tumors have rounded fusomes normally associated with the undifferentiated GSC fate. These are detected by α-HTS staining (red in **B**, white in **B'**). The nuclei of these cells counterstained with DAPI have diameters less than 10 μm (blue in **B**). (**B–B'**) bam$^{-/-}$;p53$^{-/-}$ tumors frequently exhibit disorganized fusomes detected here by α-HTS staining (red in **C**, white in **C'**, yellow arrowhead). These tumors also have many fragmented and enlarged nuclei with a diameter significantly greater than 10 μm (blue in **C**, yellow arrow). (**C**) Quantification of altered fusome structure and irregular nuclei in bam$^{-/-}$ and bam$^{-/-}$;p53$^{-/-}$ tumors. Note that in panel **C**, counts for irregular nuclei do not include micronuclei. A total of 14 cysts were assayed in bam$^{-/-}$;p53$^{-/-}$ and 8 cysts were assayed for bam$^{-/-}$. All scale bars, 10 μm.

**Table 1.** p53 status impacts expression profiles in *bam−/−* tumors

| | Downregulated by p53 | | Upregulated by 53 | |
|---|---|---|---|---|
| | Gene symbol | Fold change | Gene symbol | Fold change |
| 1 | CG31681 | 8.7 | CG31809 | −7.2 |
| 2 | CG5156 | 8.0 | CG31810 | −5.6 |
| 3 | LysX | 7.9 | CG2177 | −5.2 |
| 4 | CG31901 | 7.6 | CG7106 | −5.1 |
| 5 | CG16762 | 7.5 | CG1504 | −4.5 |
| 6 | CG32277 | 7.3 | CG15614 | −4.3 |
| 7 | CG17239 | 7.2 | unpg | −4.2 |
| 8 | CG17012 | 7.1 | CG7329 | −4.2 |
| 9 | CG9897 | 7.1 | CG15236 | −4.1 |
| 10 | Ser12 | 6.8 | CG9294 | −4.1 |
| 11 | CG2191 | 6.8 | esg | −3.9 |
| 12 | CG33258 | 6.6 | Ugt36Ba | −3.7 |
| 13 | CG18125 | 6.5 | CG14297 | −3.6 |
| 14 | CG12780 | 6.4 | CG17129 | −3.6 |
| 15 | CG4783 | 6.3 | Cyp6a14 | −3.6 |
| 16 | Cyp6a18 | 6.3 | CG5568 | −3.4 |
| 17 | CG17234 | 6.2 | CG1077 | −3.3 |
| 18 | CG18063 | 6.2 | CG11226 | −3.3 |
| 19 | CG9568 | 6.1 | CG33105 | −3.1 |
| 20 | CG32834 | 6.0 | CG3328 | −3.1 |

We performed microarray analysis on bam−/− and bam−/−;p53−/− tumors. The genes that are altered by p53 status in bam−/− tumors are recorded. Listed on the left are the top 20 genes whose abundance is directly or indirectly suppressed by p53. Listed on the right are the top 20 genes whose abundance is directly or indirectly induced by p53. The gene symbol is listed on the left and the fold change in gene expression between bam−/− and bam−/−;p53−/− tumors is listed on the right. Many genes listed here are dramatically affected when p53 is absent.

**Source data 1**. Expression features of the top 20 genes suppressed by p53. The top 20 genes that were suppressed by p53 in *bam−/−;p53−/−* tumors (see *Table 1*) were examined using GEXC (*Seita et al., 2012*) to identify enriched pathways. Using this collection we observed a mild enrichment for genes that were absent in embryos or absent in adult somatic tissues relative to all genes in the fly genome.

tissue damage (*Figure 1—figure supplement 1*). We note that stem cell specificity was not an artifact intrinsic to the biosensors, since independent reporters behaved similarly in both the female and male germline and required the wild-type Dp53 locus in both cases. Furthermore, in certain mutant backgrounds stress-induced activity restricted to GSCs was lost and non-selective p53 activation was widespread throughout the ovary (*Figure 1—figure supplement 3B'*). Therefore, despite the fact that it is present and activatable throughout the gonad, functional p53 is restricted to stem cells and their immediate progeny by specific genetic determinants.

Collectively, our work supports previous indications that there is an intimate and ancient link between p53 and stem cell biology (*Pearson and Sanchez Alvarado, 2009*). Our findings also offer rare and novel opportunities to operationally mark the stem cells in the fly germline, as visualized by p53R-GFP. This marker is distinct from conventional stem cell labels (*Deng and Lin, 1997*) since it is not constitutively expressed but, instead, represents a functional output that is conditional upon a perturbation. We further note that like all reporter systems, our p53 biosensors may not reflect the full scope of effector output regulated by this network, and activities visualized here could transmit only subsets of p53-mediated responses. Nevertheless, despite this possible limitation, our results are consistent with suggestions that stem cells may be acutely sensitive to sources of genomic instability

with a higher propensity for engaging adaptive responses relative to other cells (*Mandal et al., 2011*; *Sperka et al., 2012*). We propose that in reproductive tissues, the p53 regulatory network is either preferentially licensed in stem cells or selectively blocked outside of this compartment.

What upstream regulators might specify p53 activation in GSCs/CBs? Given that stem cells have unique properties, p53 activation in these cells might lie downstream of a novel pathway. Consistent with this idea, ATR expression, was not rate limiting for p53 activation in the germline (*Figure 1— figure supplement 3A*', *Figure 1—source data 1C*). Furthermore, unlike meiotic induction, p53 induction in GSCs/CBs was independent of the topoisomerase, Spo11 (*Figure 1—source data 1A*; *Lu et al, 2010*). Chk2 could contribute to the selective activation in stem cells seen here, but since Chk2 is also broadly expressed and functionally associated with oocyte development throughout the ovary (*Abdu et al., 2002*; *Oishi et al., 1998*) any potential role in GSCs must extend beyond a simple presence or absence of this kinase.

Our findings also imply stimulus-dependent effectors of p53 in stem cells that are not yet appreciated. For example, within detection limits, we observed no obvious connection between p53 status and apoptosis, DNA double-strand break repair, or cell cycle arrest. However, irradiated $p53^{-/-}$ GSCs were significantly delayed in the re-entry phase for cell cycle. Future studies will explore this defect and also examine progeny derived from stressed GSCs for transgenerational phenotypes that might be adaptive.

Our discovery that p53 action is coupled to hyperplasia in a non-vertebrate species was unexpected for two reasons. First, the role of this gene family as a tumor suppressor is thought to be a derived feature that evolved only in vertebrate lineages. Second, the canonical ARF/MDM2 pathway that links aberrant growth to p53 is absent outside of higher vertebrates (*Lu et al., 2009*). Surprisingly, our combined results suggest that ancient pathways linking p53 to aberrant stem cell proliferation may predate the divergence between vertebrates and invertebrates.

## Materials and methods

### Fly stocks and genetics

All fly stocks were maintained at 22–25°C on standard food media. We obtained *rad54*, *aubergine* and *cutoff* mutants: $rad54^{RU}$, $rad54^{AA}$, $aub^{HN}$, $aub^{QC}$, $cuff^{WM}$, and $cuff^{QQ}$ from T Schupbach (Princeton University, Princeton, NJ, USA); *c587*-GAL4, UAS-*dpp*, UAS- $Lsd1^{KD}$ (*Eliazer et al., 2011*), homozygous viable allele of *bamΔ86* (*McKearin and Spradling, 1990*; *McKearin and Ohlstein, 1995*), *nanos*-GAL4VP16, and UASp-*tkvCA* (*Chen and McKearin, 2003*) have been described previously. All other stocks were obtained from Bloomington Stock Center (Indiana University, Bloomington, IN, USA). The Dp53 rescue strain was engineered by φC31 integration of a 20-kb genomic fragment BAC containing the Dp53 locus into an attP site on the X chromosome of the PBac{y + -attP-9A}VK00006 line (Bloomington #9726). The parent BAC CH322-15D03 was obtained from the P[acman] resource library (*Venken et al., 2009*) and Rainbow Transgenic Flies performed the injection and screening for recombinants. The I-SceI endonuclease strain was generated by K Galindo (*Galindo et al., 2009*), which was crossed to p53R-GFPnls(STI150); HS-(70Flp)(70 I- Sce I)/TM6 for heat-inducible I-SceI endonuclease expression. Adult females were fattened for 2–3 days after eclosion and then subjected to heat shock in a circulating water bath at 37°C for 90 min and repeated for three consecutive days. 24 hr after the last heat shock, ovaries were dissected for immunostaining. For forced proliferation assays, two GAL4 lines were used: *nanos*-GAL4VP16 was used to achieve overexpression in the germline with UAS constructs for Ras$^{V12}$, CyclinE, and Thickveins (*Rorth, 1998*). *c587*-GAL4 was used to achieve overexpression of UAS constructs of *Dpp* or *Lsd1-RNAi* in the somatic cells of the ovariole tip (*Song et al., 2004*). For cyclinE overexpression, stocks were maintained at 25°C and female virgins were collected upon eclosion, shifted to 29°C for 4–5 days then subjected to immunostaining. For the Ras$^{V12}$ studies, female virgins were shifted to 29°C for 1 day and then shifted down to 25°C for 3 days prior to immunostaining. The Gal4-UAS system (adapted from yeast) often produces optimal expression at temperatures higher than 25°C. Since the UAS-Rasv12 and UAS-CyclinE constructs were not optimized for expression in the germline we applied these temperature shifts to produce more penetrant phenotypes.

### Irradiation assay

Well-fed flies were exposed to ionizing radiation using a Cs-137 Mark 1-68A irradiator (JL Shepherd & Associates, San Ferando, CA, USA) at a dose of 4 krad unless otherwise noted. When irradiating several genotypes, each genotype was placed in an individual vial, and all vials were exposed to IR at

the same time on a rotating turntable inside the irradiator. For visualizing reporter activation after IR, flies were dissected 24 hr post-IR to allow for stable GFP expression.

## Immunostaining of fly tissue

3- to 5-days-old well-fed females were dissected in PBS and fixed in 4% EM-grade formaldehyde (Polysciences, Warrington, PA) diluted in PBS-0.1% tween-20, with three times the volume of heptane. After washing, tissues were blocked in 1.5% BSA, then incubated with primary antibodies at 4°C overnight. Antibodies used: rabbit α-GFP (Invitrogen, Carlsbad, CA); rabbit α-pH2Av (kindly provided by K McKim with specific staining protocols), rabbit α-cleaved caspase 3 (Asp175) (Cell Signaling, Danvers, MA); mouse α-Armadillo, mouse α-BrdU (Sigma, St. Louis, MO), mouse α-HTS clone 1B1 (Developmental Studies Hybridoma Bank, Iowa City, IA), and rat α-Vasa (Developmental Studies Hybridoma Bank). For fluorescence visualization, Alexa-488, 568 (Invitrogen), and DyLight 649 (Jackson ImmunoResearch, West Grove, PA) secondary antibodies were used and 0.1 µg/ml of DAPI (Invitrogen) for DNA staining was added in the first wash step. After three washes, ovaries were further hand dissected and mounted in VECTASHIELD (Vector Laboratories, Burlingame, CA) for microscopy imaging. For validating stimulus-dependent p53 action as visualized by the reporters, we routinely confirmed absence of GFP expression using flies null for Dp53. We note that p53R-GFPnls shows constitutive expression independent of p53 in a subset of gut cells and in the region of the testis containing elongated spermatids, reflecting position effects upon this transgene.

## Fertility tracking and proliferative arrest assay

In fertility assays, two *p53* null alleles, 238H (ns) and 5A-1-4 (k1) were used in trans-combination to reduce genetic background influences. Two wild-type strains, *yw* and *w1118* were used for comparison. p53 rescue transgenes were tested in a transheterozygous p53$^{-/-}$ background (A1; ns/k1 and A2; ns/k1) to exclude contributions from background modifiers. 5- to 7-day-old females were irradiated at desired doses (11.5 krad for *Figure 4B* and 9 krad for *Figure 4—figure supplement 3*) and fertility was tracked over time in groups. Each group contained 10 females and five unirradiated wild-type Canton-S males. The animals were transferred to a new vial at designated time points, and fertility was scored by the presence of larvae 10 days after the parents were removed. Each trial contained 2 to 15 replicates per genotype. For *Figure 4B* percentages of fertile samples are plotted based on five trials. In the proliferative arrest assay, ovaries were dissected and immersed in Grace's media containing BrdU (10 µM) for 1 hr at room temperature. After fixation, ovaries were treated with 2N HCl for 30 min then 100 mM of borax was added for 2 min to neutralize the pH. Tissues were then processed for blocking and regular immunostaining.

## Statistical analysis

For all statistical analysis, data were placed into GraphPad Prism software. For statistics on the IR and Isce-I reporter activation (*Figure 1G*, *Figure 1—source data 1*), one-way ANOVA test was performed on all genotypes with a Tukey's Multiple Comparison post-test. Reporter activation in *aubergine*, *cutoff*, and *rad54* mutants (*Figure 3*, *Figure 3—source data 1*) was analyzed using a two-tailed unpaired t-test comparing the transheterozygous mutant to the heterozygous control. The same analysis was carried out for region 3 and stage 2–8 (*Figure 3—figure supplement 1*, *Figure 3—source data 1*). For statistical analysis on fertility and BrdU incorporation assays (*Figure 4*), one-way ANOVA test was performed for each time point with a Dunnett post-test in which p53$^{-/-}$ data was the control. For cleaved-caspase 3 analysis (*Figure 4—figure supplement 1*), the data was analyzed using a two-tailed unpaired *t* test. In cases where replicates produce identical values incompatible with the prism two-tailed unpaired t-test tool, one value was negligibly revised to enable computation by this software (e.g., when both values were 0, one value was changed to 1.0$^{e-12}$).

## Microarray and Gene Expression Commons (GEXC) analysis

About 200 ovaries from bam or bamp53 adult females were dissected in batches and pooled together to extract total RNA using Trizol (Invitrogen). After verifying RNA integrity using Bioanalyzer (2100; Agilent), whole-genome expression of each genotype was analyzed using Affymetrix Drosophila Genome 2.0 Array at UTSW Genomics & Microarray core facility. Microarray data sets were uploaded to Gene Expression Commons (https://gexc.stanford.edu) and analyzed with 17 other public available data sets. In Gene Expression Commons, raw microarray data is individually normalized against a large-scale common reference (for *Drosophila* genome, n = 2687 as of Nov 2013), mapped onto the

probeset meta profile. This strategy enables profiling of absolute expression levels of all genes on the microarray, instead of conventional methods where differences in gene expression are compared only between samples within an individual experiment (*Seita et al., 2012*).

## Acknowledgements

We thank the following for kindly providing materials: Dr Trudi Schupbach for *rad54*, *aubergine* and *cutoff* mutants, Dr Dennis McKearin for bam, nanos-GAL4VP16, and UASp-TkvCA strains, and Dr Kim McKim for α-pH2Av and staining protocols. We thank Dr Susan Eliazer for providing unpublished strains and information. We greatly appreciate Ashley Aguilar for help with dissections and immuno-histochemistry and Jessica Alatorre for maintaining fly stocks.

## Additional information

### Funding

| Funder | Grant reference number | Author |
| --- | --- | --- |
| CPRIT | RP110076 | John M Abrams |
| Ellison Foundation | AG-SS-2743-11 | John M Abrams |
| National Institute of General Medical Sciences | R01GM072124 | John M Abrams |
| Welch Foundation | I-1727 | John M Abrams |
| Genetics Training Grant | 5T32GM083831 | Annika Wylie |
| National Institute of General Medical Sciences | GM086647 | Michael Buszczak |

The funders had no role in study design, data collection and interpretation, or the decision to submit the work for publication.

### Author contributions

AW, W-JL, AD'B, Conception and design, Acquisition of data, Analysis and interpretation of data, Drafting or revising the article; MB, Conception and design, Analysis and interpretation of data, Drafting or revising the article, Contributed unpublished essential data or reagents; JMA, Conception and design, Analysis and interpretation of data, Drafting or revising the article

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
