## [Decision Letter]

Thank you for sending your work entitled “p53 activity is selectively licensed in the *Drosophila* stem cell compartment” for consideration at *eLife*. Your article has been evaluated by a Senior editor, a Reviewing editor, and 3 reviewers.

The Reviewing editor and the three reviewers discussed their comments and the Reviewing editor has assembled the following comments that require your response before we can make a determination on the suitability of this work for publication in *eLife*.

The referees feel that your findings are intriguing and potentially of sufficient interest for the broad audience of *eLife* because of the provocative finding that p53 may be activated exclusively in germline stem cells after myriad forms of stress. Yet, it was felt that your findings as described are somewhat preliminary and that further details on the roles of p53 in stem cells need to be provided. The reviewers also felt that the text has to be clarified in several instances. Please consider the following points: 

1) The authors observe a phenotype of p53 mutants on recovery from sterility and re-entry into the cell cycle after irradiation. However, it's not clear if the delay in cell cycle re-entry (which is quite modest, especially after 72 hours after irradiation) causes the infertility. For example, at higher doses of irradiation (Figure 4—figure supplement 3), wild type flies did not become sterile and p53 mutants showed a reduced fertility (but not complete infertility). Does this correlate with an accelerated re-entry into the cell cycle (when compared with lower doses of irradiation)? 

2) The role of apoptosis in these phenotypes is also insufficiently explored. Staining with cleaved caspase 3 was performed but did not implicate apoptosis. And is it clear whether inhibition of apoptosis would produce any phenotype in these germaria? Further, what is measured is caspase 3 activation per germarium, which doesn't take into account the number of GSCs or CBs per germarium. C3 is a good marker for induction of apoptosis but doesn't indicate how many cells may have already died and been cleared. 

3) What is the role of p53 in stem cell tumors? The authors show activation of p53 in different tumor models. But what is the phenotype of these tumors when they are also mutant for p53? Would these tumors grow even bigger? Also, in the different tumor models that they use, there seems to be a different pattern of p53 activation. With Ras overexpression they observe a pattern very similar to the one they see after irradiation. However, the rest of oncogenic stresses (*bam*^*-/-*^*, nanos*>*CyE, nanos*>*TkvCA, c587*>DPP and *c587*>*Lsd1KD*) show a totally different pattern, and in many cases it looks that p53 activation is not restricted to stem cells. In the case of *bam*^*-/-*^ they show that there is an expansion of the stem cell population, and that cells that activate p53 also express the germline marker Vasa. Is that also the case for the rest of them? Or otherwise, what would be the explanation for the different pattern of p53 activation? 

4) What is the evidence that the reporter transgene, which contains an enhancer from the known p53 target gene reaper fused to GFP, actually reports all activation of p53? That is, is this reaper enhancer activated in response to p53 activity in all cell types and under all conditions that p53 is “activated”? Without such knowledge, which is admittedly nontrivial to obtain, the main conclusion that “adult *Drosophila* exposed to genotoxic stress or genome destabilizers selectively activated p53 in GSCs and their immediate progeny” cannot be made. The data in Figure 1—figure supplement 3 may actually address this point somewhat. These data show that the reporter can be activated in non-GSCs of the ovary when ATR is mutated (both follicle cells and nurse cells it appears, although a better image is needed to make this call). Is this activation p53 dependent? If so, then the restriction of reporter activity to GSCs in response to IR is ATR dependent, which would be somewhat counter intuitive since ATR normally activates p53. What is the activity of the reporter in ATR mutants in the absence of IR? In the absence of additional experiments, a more prudent or expansive interpretation of the current data seems warranted. 

5) There are also problems with the presentation and description of the results. As some examples: – In vertebrates there is strong evidence for a role of p53 as a master regulation of stem cell proliferation and differentiation. Here it appears that they don't see a significant change of stem cell numbers in p53 mutants, but this is never clearly stated. 

– Often an experimental result is given but only cursory information is given to assist the reader in interpreting that result. Some of this is because explanations that belong in the main text are only found in supplementary figure legends. 

– Another example occurs with the introduction of the *cuff* and *aub* mutations, where the phenotype of these mutants in ovaries is not described other than to say they are involved in retrotransposon silencing. 

– There are also inconsistencies in some of their results (i.e., biosensor activation outside of the germline) that are not addressed except for the suggestion they result from lack of penetrance or GFP perdurance, etc. How often are these deviations from their model observed?

---

## [Author Response]

*1) The authors observe a phenotype of p53 mutants on recovery from sterility and re-entry into the cell cycle after irradiation. However, it's not clear if the delay in cell cycle re-entry (which is quite modest, especially after 72 hours after irradiation) causes the infertility. For example, at higher doses of irradiation (*Figure 4—figure supplement 3*), wild type flies did not become sterile and p53 mutants showed a reduced fertility (but not complete infertility). Does this correlate with an accelerated re-entry into the cell cycle (when compared with lower doses of irradiation)*?

Good suggestion. Transient sterility and delayed re-entry into the cell cycle are phenotypes that we clearly link to p53^-^ status but whether the relationship between these defects is direct or indirect is an open question. The studies suggested here provided a way to extend our correlations between defective cell cycle and reduced fertility. Our revision includes the suggested Brdu labeling experiments on samples at informative time points challenged as in original Figure S3 (now Figure 4—figure supplement 3). Also, please note that the experiments in current Figure 4—figure supplement 3 applied lower irradiation doses when compared to original Figure 3 (now Figure 4). In retrospect, we realize that our presentation was confusing and the revised manuscript clarifies these issues.

At an IR dose of 4 krad (current Figure 4) we observe a modest delay in cell cycle re- entry but no effect on fertility. At an IR dose of 11.5 krad, WT flies transiently become infertile while p53-/- flies become permanently infertile (current Figure 4). In our revised submission we performed Brdu labeling experiments at 4hrs, 2 days, and 7 days post-irradiation on an intermediate dose where WT flies remain fertile and p53^-/-^ flies show decreased fertility (current Figure 4—figure supplement 3). Here we observe that p53^-/-^ flies have a persisting delay in cell cycle re-entry, even over 7 days. We have added this new data to the supplement (current Figure 4—figure supplement 3) and have expanded on these observations in the results section.

*2) The role of apoptosis in these phenotypes is also insufficiently explored. Staining with cleaved caspase 3 was performed but did not implicate apoptosis. And is it clear whether inhibition of apoptosis would produce any phenotype in these germaria? Further, what is measured is caspase 3 activation per germarium, which doesn't take into account the number of GSCs or CBs per germarium. C3 is a good marker for induction of apoptosis but doesn't indicate how many cells may have already died and been cleared*.

p53 genes often control damage-induced apoptotic responses and examining cell death in these germaria was strongly justified. We were surprised that no cell death role was indicated but we also note here that defective apoptosis does not appear to adequately explain tumor suppression by p53 in mice (see [41], Martins 2006). In our experience, immunohistochemistry with the CC3 antibody is a reliable and robust marker for apoptotic cells. However, like all cell death markers, labeling is transient because the affected cells and their corpses ultimately disappear. For this reason – and to insure a comprehensive examination for possible apoptogenic defects – we completed a well- resolved time course at 2hr, 4hr, 8hr and 24hr post-irradiation (see current Figure 4—figure supplement 1). Because we obtained a negative result from these exhaustive studies, we did not pursue this point further. Furthermore, tests could be conducted to ask whether inhibition of apoptosis produces a phenotype in the germarium as suggested – however, the results would not help us deconstruct p53 action in the context of stem cells, since a role for cell death was not indicated. Therefore, in our view, the results could be of interest in their own right but would not advance the story we are driving at here.

*3) What is the role of p53 in stem cell tumors? The authors show activation of p53 in different tumor models. But what is the phenotype of these tumors when they are also mutant for p53? Would these tumors grow even bigger*?

Good questions. We carefully examined these tumors to address precisely these points and we included new experiments establishing that p53 does have a role. We did not observe consistent size differences but we did find characteristic affects on cytology and altered gene expression profiles governed by p53 status. These new findings are included in the revision (see Figure 6, Table 1, and [Supplementary-material SD5-data]).

*Also, in the different tumor models that they use, there seems to be a different pattern of p53 activation. With Ras overexpression they observe a pattern very similar to the one they see after irradiation. However, the rest of oncogenic stresses (bam*^*-/-*^*, nanos>CyE, nanos>TkvCA, c587>DPP and c587>Lsd1KD) show a totally different pattern, and in many cases it looks that p53 activation is not restricted to stem cells. In the case of bam*^*-/-*^
*they show that there is an expansion of the stem cell population, and that cells that activate p53 also express the germline marker Vasa. Is that also the case for the rest of them? Or otherwise, what would be the explanation for the different pattern of p53 activation*?

We find it remarkable that p53 reporter is consistently activated in fly tumor models. The dissimilarities in activation patterns seen here most likely trace to differences in the underlying histology and/or differentially deranged signaling in the hyperplasias themselves. In the revised submission we discuss these differences and consider possible explanations where appropriate. The dissimilarities could be related to signaling dynamics, response dynamics, cell cycle dynamics etc. Technical sources of variation linked to subtle differences in the action of the UAS-GAL4 driver system and/or non-uniform accumulation of the oncogenic product are also possible. Please note that none of these considerations detract from the compelling positive findings that we are documenting here. In each of the models tested here (original Figure 4, now Figure 5) supernumerary and/or ectopic germline stem cells are already very well documented (Chen 2003; Elazier 2011; Xie 1998; Song 2004). Furthermore, we are confident that p53-activated cells are indeed supernumerary germline cells and, in many cases, a rounded fusome is visible in GFP+ cells. In response to this concern, we revised current Figure 5 so that it now includes higher magnification insets in Figure 5. These images show that p53R-GFP+ cells also have a rounded fusome, which marks germline stem cells. We also substituted panel F in Figure 5 with an improved DAPI stained confocal image.

*4) What is the evidence that the reporter transgene, which contains an enhancer from the known p53 target gene reaper fused to GFP, actually reports all activation of p53? That is, is this reaper enhancer activated in response to p53 activity in all cell types and under all conditions that p53 is “activated”? Without such knowledge, which is admittedly nontrivial to obtain, the main conclusion that “adult Drosophila exposed to genotoxic stress or genome destabilizers selectively activated p53 in GSCs and their immediate progeny” cannot be made*.

The reviewers suggest that our interpretation is unreasonable because our sensors might not be perfect. We strongly disagree with this argument. The comment cites limitations that are inherent to virtually all tools that are used in the cell sciences. In fact, the same concern can be raised for any reporter, any biosensor, any antibody, any phospho-specific antibodies, etc. While they may not reflect the entire scope of output from the p53 regulatory network, our reporters are advantageous because they are validated functional sensors for p53 action in vivo. Furthermore, it is not apparent how one might produce idealized sensors that are immune to this caveat. In the revised text (Discussion, paragraph 2), we explicitly consider this limitation.

*The data in*
Figure 1—figure supplement 3
*may actually address this point somewhat. These data show that the reporter can be activated in non-GSCs of the ovary when ATR is mutated (both follicle cells and nurse cells it appears, although a better image is needed to make this call). Is this activation p53 dependent? If so, then the restriction of reporter activity to GSCs in response to IR is ATR dependent, which would be somewhat counter intuitive since ATR normally activates p53*.

The relationship between ATR and the p53 network had not been previously examined in flies. Also, given that ATR is not universally required for p53 in mammalian systems (see Nghiem, 2002) the findings are not unexpected. The fact that we see ectopic activation is actually consistent with the probability that ATR exerts constitutive genome stabilizing activities. Where possible, the biosensors are routinely validated by testing them in the p53^-^ background. However, we are unable to recover viable animals that are simultaneously homozygous for the biosensor, homozygous for ATR^-^ and also homozygous for p53^-^. Ectopic induction of the biosensor here (and in other mutants) raises intriguing implications. But, for purposes of discussion here, the findings have strong merit because they also exclude trivial explanations for the restricted stem cell activity that we observe in the wild type context. For example, it was possible that p53 was simply confined to stem cells or was somehow inert in other cells. It was also possible, for technical reasons, that the biosensor was somehow unable function in somatic cells of the ovary. Our rationale for including these results was twofold: first, we felt it was important to demonstrate that p53 and the biosensor are normally present and ‘activatable’ throughout the ovary; second, we felt it was useful to include additional examples showing that stem cell selectivity is under genetic control.

*What is the activity of the reporter in ATR mutants in the absence of IR? In the absence of additional experiments, a more prudent or expansive interpretation of the current data seems warranted*.

The revised manuscript adds a new graph (current Figure 1—figure supplement 3) that quantifies reporter activity in GSCs and in follicle cells of ATR mutants. This information will enable readers to appreciate reporter status in the presence and absence of IR.

*5) There are also problems with the presentation and description of the results. As some examples*:

We extensively revised the text to clarify our presentation and our interpretations as requested.

*– In vertebrates there is strong evidence for a role of p53 as a master regulation of stem cell proliferation and differentiation. Here it appears that they don't see a significant change of stem cell numbers in p53 mutants, but this is never clearly stated*.

Good point. The revision adds a section in the results where this topic is explicitly discussed. Modestly altered stem cell pools in certain tissues of p53 mutant mice have been reported to be affected by age (Dumble, 2007). However, to our knowledge, stem cell pools in p53 mutant mice are not universally elevated (Reviewed in Solozobova and Blattner, 2011). Instead, it seems that the master regulation concept mostly derives the effects of p53 upon iPS cell production (Krizhanovsky, 2009) and there are alternative interpretations for the this effect (e.g., prevention of apoptosis).

*– Often an experimental result is given but only cursory information is given to assist the reader in interpreting that result. Some of this is because explanations that belong in the main text are only found in supplementary figure legends*.

In retrospect, we agree. This is a valid criticism. The revised manuscript comprehensively addresses this through improved transitions and additional background information where needed. For example, we divided the ovary and testis data in original Figure 1 into two figures (current Figures 1 and 2). We also moved the description of the p53 biosensors from the supplement to current Figure 1.

*– Another example occurs with the introduction of the cuff and aub mutations, where the phenotype of these mutants in ovaries is not described other than to say they are involved in retrotransposon silencing*.

The revised manuscript includes additional background and context on these mutants.

*– There are also inconsistencies in some of their results (i.e., biosensor activation outside of the germline) that are not addressed except for the suggestion they result from lack of penetrance or GFP perdurance, etc. How often are these deviations from their model observed*?

To address this concern, we have improved our presentation of these issues. We emphasize that our ‘licensing’ model refers to the stimulus-dependent action of p53 in a wild type setting. When wild type ovaries are radiation challenged, functional p53 activation is seen only in the stem cell and their immediate progeny – under these conditions, the p53 biosensor is never observed outside of the germ stem cell compartment. This could reflect independent activation in stem cells and their immediate progeny or, alternatively, the GFP signal in cystoblasts could reflect perdurance of GFP from parental stem cells (we favor this for reasons discussed in the text). However, in the ovaries of mutants that affect genome stability, we do observe reporter activation outside of the germline and this observation provides us with important insights. In these cases, we propose that the corresponding genes normally function in ways that ultimately restrict widespread activation. In mutants that produce oncogenic stress, p53 is more broadly activated but the ectopic activation here still occurs within germline cells.